# Partial substitution of chemical fertilizer by organic fertilizer increases yield, quality and nitrogen utilization of *Dioscorea polystachya*

Chao Sun[1], Hua Zheng[2], Shuxia He[1], Qing Zhao[3], Yuxi Liu[4], Hai Liu[1] *

**1** School of Management Science and Engineering, Guizhou University of Finance and Economics, Guiyang, Guizhou, China, **2** Guizhou Tobacco Company, China National Tobacco Corporation, Guiyang, Guizhou, China, **3** Academic Affairs Office, Guizhou University of Finance and Economics, Guiyang, Guizhou, China, **4** School of Literature, Guizhou University of Finance and Economics, Guiyang, Guizhou, China

* liuhai200321@163.com

**Data Availability Statement:** All relevant data are within the manuscript and its Supporting Information files

## Abstract

This field experiment aimed to investigate the effects of different ratios of organic and inorganic fertilizers with maintaining equal nitrogen application rates on the yield, quality, and nitrogen uptake efficiency of *Dioscorea polystachya* (yam). Six treatments were set, including a control without fertilizer (CK), sole application of chemical fertilizer (CF), sole application of organic fertilizer (OM), 25% organic fertilizer + 75% chemical fertilizer (25%OM + 75%CF), 50% organic fertilizer + 50% chemical fertilizer (50%OM + 50%CF), and 75% organic fertilizer + 25% chemical fertilizer (75%OM + 25%CF). The experiment followed a randomized complete block design with three replications. Various yield parameters, morphology, quality indicators, and nitrogen utilization were analyzed to assess the differences among treatments. The results indicated that all fertilizer treatments significantly increased the yield, morphology, quality indicators, and nitrogen utilization efficiency compared to the control. Specifically, 25%OM + 75%CF achieved the highest yield of 31.96 t hm$^{-2}$, which was not significantly different from CF (30.18 t hm$^{-2}$). 25%OM + 75%CF exhibited the highest values at 69.23 cm in tuber length and 75.86% in commodity rate, 3.14% and 1.57% higher than CF respectively. Tuber thickness and fresh weight of 25%OM + 75%CF showed no significant differences from CF, while OM and 50%OM+50%CF exhibited varying degrees of reduction compared to CF. Applying fertilizer significantly enhanced total sugar, starch, crude protein, total amino acid, and ash contents of *D. polystachya* (except ash content between CK and OM). Applying organic fertilizer increased the total sugar, starch, crude protein, total amino acid, and ash contents in varying degrees when compared with CF. The treatment with 25%OM+75%CF exhibited the highest increases of 6.31%, 3.78%, 18.40%, 29.70%, and 10%, respectively. Nitrogen content in different plant parts followed the sequence of tuber > leaves > stems > aerial stem, with the highest nitrogen accumulation observed in 25%OM + 75%CF treatment. Nitrogen harvest index did not show significant differences among treatments, fluctuating between 0.69 and 0.74. The nitrogen apparent utilization efficiency was highest in 25%OM + 75%CF (9.89%), followed by CF (9.09%), both significantly higher than OM (5.32%) and 50%OM + 50%CF (6.69%). The nitrogen agronomic efficiency varied significantly among treatments, with 25%OM + 75%CF

**Funding:** This research was funded by the Guizhou Planning Office of Philosophy and Social Science (Grant No. 21GZYB61). The funders had no role in study design, data collection and analysis, decision to publish, or preparation of the manuscript.

**Competing interests:** The authors have declared that no competing interests exist.

(33.93 kg kg$^{-1}$) being the highest, followed by CF (29.68 kg kg$^{-1}$), 50%OM + 50%CF (21.82 kg kg$^{-1}$), and OM (11.85 kg kg$^{-1}$). Nitrogen partial factor productivity was highest in 25%OM + 75%CF treatment (76.37 kg kg$^{-1}$), followed by CF (72.11 kg kg$^{-1}$), both significantly higher than 50%OM + 50%CF (64.25 kg kg$^{-1}$) and OM (54.29 kg kg$^{-1}$), with OM exhibiting significantly lower values compared to other treatments. In conclusion, the combined application of organic and inorganic fertilizers can effectively enhance the yield, quality, and nitrogen utilization efficiency of *D. polystachya*. Particularly, the treatment with 25% organic fertilizer and 75% chemical fertilizer showed the most promising results.

## Introduction

Yam (*Dioscorea polystachya*) is a perennial herbaceous plant that holds significant economic value as a vital crop in China [1]. It is highly esteemed for its nutritional and medicinal properties, serving as a crucial source of income for farmers [2]. However, the extensive growth cycle and high nutrient requirements of yam necessitate significant fertilizer application. Unfortunately, in the pursuit of enhanced yields, farmers often resort to excessive utilization of chemical fertilizers, resulting in increased production costs and adverse environmental impacts [3, 4]. Relevant research findings have demonstrated that a substantial proportion of unused fertilizers persist in the soil, leading to adverse consequences such as soil compaction and salinization [5]. Additionally, these unused fertilizers can infiltrate the atmosphere and water bodies through various pathways like ammonia volatilization, nitrification, denitrification, surface runoff, and leaching. As a result, this contributes to exacerbating atmospheric pollution and eutrophication of water bodies [6, 7]. Therefore, it imperative to explore comprehensive models of agricultural development models and nutrient management strategies that effectively meet China's food demands while simultaneously mitigating the strain on resources and the environment [8]. This approach is crucial for achieving sustainable and "green" agricultural development in China [9].

Researchers have discovered that the exclusive utilization of organic fertilizer alone can result in a significant decrease in crop yield. Under low-input conditions, the application of organic fertilizer alone led to a reduction in crop yield ranging from 24% to 54%, compared to the application of chemical fertilizer alone or the combined application of organic and inorganic fertilizers [10]. Moreover, a six-year field trial conducted in Sweden demonstrated that relying solely on manure as a fertilizer resulted in substantial declines of 47% and 82% in barley and potato yields respectively [11]. Numerous studies have underscored the advantages of combining organic and chemical fertilizers, as this approach has the potential to enhance crop yield and improve soil quality. This combined application strategy exhibits significant promise for amplifying agricultural production, optimizing efficiency, and reducing fertilizer usage in China..

In Guizhou province, the predominant practice has been the heavily reliant on chemical fertilizers, with limited utilization of organic fertilizers. This study aimed to evaluate the impact of substituting different proportions of organic fertilizer for chemical fertilizers on yam yield, quality, morphology, and nitrogen utilization efficiency. The findings from this research endeavor aim to provide valuable guidance towards reducing reliance on chemical fertilizers, promoting appropriate application of organic alternatives, and effectively enhancing yam productivity in the Guizhou yam planting region."

## Materials and methods

### Overview of the study site

The experimental site was located in the University Town of Huaxi District, Guiyang City, Guizhou Province, China (26˚ 22′N, 106˚ 38′E). This location is situated within the hilly terrain of central Guizhou Province, characterized by an average altitude of 1030 m. The region experiences a subtropical monsoon climate, with an average annual temperature of 15.3˚C and approximately 1350 hours of sunshine per year. The frost-free period spans around 270 days while annual rainfall typically ranges between 1100–1200 mm.

The experimental yellow soil (Eutric Regosol, UAO Soil Taxonomic System) was sampled from the 0–20 cm soil layer prior to the experiment and subsequently analyzed using the methods listed below. The soil originated from Triassic limestone and sand shale deposits, exhibiting a sandy loam texture with a pH of 5.74. It contained 15.41 g kg$^{-1}$ of organic matter (OM), 0.82 g kg$^{-1}$ of total nitrogen (TN), 0.41 g kg$^{-1}$ of total phosphorus (TP), 11.89 g kg$^{-1}$ of total potassium (TK), 75.18 mg kg$^{-1}$ of N hydrolyzed by NaOH solution (AN), 16.8 mg kg$^{-1}$ of Olsen P (AP), and 109.6 mg kg$^{-1}$ of K exchangeable with $CH_3COONH_4$ solution (AK).

### Test material

1. Yam variety: The test variety utilized in this study was Anshun Yam, a locally adapted variety derived from the domestication and natural selection of wild yam plants in the region. This variety has undergone long-term domestication and natural selection. Seeds of Anshun Yam were procured from Huaxi District Seed Company, situated in Guiyang City, Guizhou Province, China.

2. Fertilizers: This vermicompost, derived from the digestion of decomposed cow dung with earthworms, had a nutrient composition of N-$P_2O_5$-$K_2O$ = 2.01–2.41–1.03 and an organic matter content of 36%. The vermicompost was purchased from Guizhou Jilong Ecological Technology Co., LTD. Additionally, synthetic fertilizers were used, including urea (N 46%), compound fertilizer (N-$P_2O_5$-$K_2O$ = 15-15-15), superphosphate ($P_2O_5$ 16%), potassium sulfate ($K_2O$ 50%), and diammonium hydrogen phosphate (N 18%, $P_2O_5$ 46%). All of these synthetic fertilizers were procured from a local agricultural materials company.

### Experimental design

The experiment was conducted over a two-year period, from 2020 to 2021, to evaluate the effects of different treatments on yam cultivation. Six treatments were established:

1. Control (CK): No fertilization was applied;

2. Chemical fertilizer (CF): Sole application of conventional chemical fertilizer;

3. Organic fertilizer (OM): Sole application of organic fertilizer;

4. 25%OM+75%CF: 25% organic fertilizer + 75% chemical fertilizer;

5. 50%OM+50%CF: 50% organic fertilizer + 50% chemical fertilizer;

6. 75%OM+25%CF: 75% organic fertilizer + 25% chemical fertilizer.

The experiment was conducted using a randomized complete block design with three replicates. Each experimental plot measured 7.5 m × 3.6 m in size. Within each plot, six rows were arranged with a row spacing of 30 cm × 60 cm. A total of 180 yam plants (equivalent to 55,560

plants $hm^{-2}$) were planted in each plot, while maintaining a walkway width of 0.3 m between the plots. Protective rows were established surrounding the experimental plots to minimize any potential edge effects.

Yam setts were planted using the trench planting method (trench depth 15 cm) on March 12 in both 2020 and 2021. Prior to planting, a basal application of fertilizers was conducted, followed by ridging after planting (ridge height 15 cm). Staking was performed before sprouting, and trailing operations were consistently carried out by training newly sprouted yam vines to the stakes. Top dressing was applied approximately two months after planting. The field management practices adhered to local high-quality cultivation technical regulations. All fertilizing treatments employed identical application rates of N at 418.5 kg $ha^{-1}$, $P_2O_5$ at 544.5 kg $ha^{-1}$, and $K_2O$ at 712.5 kg $ha^{-1}$. CF represented the local conventional fertilizer treatment consisting of a base fertilizer of compound fertilizer at a rate of 2250 kg $ha^{-1}$ along with top-dressing fertilizers comprising potassium sulfate at a rate of 750.0 kg $ha^{-1}$ and ammonium dihydrogen phosphate at a rate of 450.0 kg $ha^{-1}$. Urea, calcium superphosphate, and potassium sulfate were used as supplements for any nitrogen deficiency in organic fertilizer applications. The basal application exclusively utilized organic fertilizer and calcium superphosphate while urea, potassium sulfate,and ammonium dihydrogen phosphate were applied as basal application with 60% of their total amount; the remaining 40% was used for topdressing. Further details regarding the type and amount of fertilizers are provided in Table 1."

## Methods of index determination

The yam was harvested during Nov. 21- Nov. 30, plot yield was weighed individually, soil ($0 \sim 20$ cm in depth) and yam tuber samples were collected through five-point sampling method in each plot separately. The measuring items and methods were listed below.

**Soil properties.** The soil samples collected from each plot were air-dried for determining the following indicators. The soil pH (soil:water = 1:10) was measured with a pH metre (FE20, Mettler Toledo Co., Ltd., Shanghai, China), and OM was determined by the $K_2Cr_2O_7$ oxidation method. After digested with $H_2SO_4$-$H_2O_2$, the soil TN, TP, and TK in the digests were respectively determined by methods of Kjeldahl distillation, molybdenum blue colorimetry and flame photospectrometry [12]. For AN determination, the soil samples were evenly mixed with $FeSO_4·7H_2O$-$Ag_2SO_4$ and added with 1 N NaOH solution (soil: solution = 1:5, weight: volume), after incubated at 40°C for 24 h, HCl titration was employed to determine $NH_3$ collected in 2% $H_3BO_3$ solution. 0.5 M $NaHCO_3$ (soil:water = 1:10) was prepared for AP extraction, which was then measured using the colorimetric molybdenum-blue method. AK was

Table 1. Fertilization schemes of different treatments (kg $ha^{-1}$).

| Treatments | Base fertilizer | | | | | | Topdressing fertilizer | | |
|---|---|---|---|---|---|---|---|---|---|
| | Compound fertilizer | Urea | Calcium superphosphate | Potassium sulfate | Ammonium dihydrogen phosphate | Organic fertilizer | Urea | Potassium sulfate | Ammonium dihydrogen phosphate |
| CK | 0 | 0 | 0 | 0 | 0 | 0 | 0 | 0 | 0 |
| CF | 2250 | 0 | 0 | 0 | 0 | 0 | 0 | 750.0 | 450.0 |
| OM | 0 | 0 | 267.2 | 597.6 | 0 | 20820 | 0 | 398.4 | 0 |
| 25%OM+75% CF | 0 | 409.4 | 2619 | 790.7 | 0 | 5205 | 273.0 | 527.1 | 0 |
| 50%OM+50% CF | 0 | 273.0 | 1835 | 726.3 | 0 | 10410 | 182.0 | 484.2 | 0 |
| 75%OM+25% CF | 0 | 136.5 | 1051 | 662.0 | 0 | 15615 | 91.05 | 441.3 | 0 |

extracted with 1 M $NH_4Ac$ and determined with a flame photometer (Shanghai Analysis Instrument Co., Ltd., China).

**Yam yield.** Following the sampling yam tubers were harvested separately per plot. The average yield of three replicate plots of each treatment was weighed and calculated to determined the overall yield.

**Tuber morphology.** Ten randomly selected yam plants were chosen from each plot to assess tuber morphology indices, encompassing measurements of tuber length and thickness.

**Yam quality assessment.** Fresh and intact yam tubers of medium length were chosen from each plot to determination of quality indices. The total ash content, soluble sugar content, starch content, and soluble protein content were determined respectively by the colorimetric analysis method [13].

**Nutrient content.** For nutrient analysis, five randomly selected yam plants were conducted from each plot. The stems, leaves, rhizomes, and tubers were individually weighing. The green parts were heated at 105˚C for 30 minutes to deactivate enzymes, and then dried at 65˚C to constant weight. The dry weight was recorded, and the samples were crushed for further analysis. After a deboiling process using concentrated sulfuric acid and hydrogen peroxide, the total nitrogen content (Kjeldahl method), total phosphorus content (vanadium-molybdenum yellow colorimetric method), and total potassium content (flame photometer) were determined for each plant part [14].

Relevant calculation formulas:

Plant nitrogen accumulation (kg ha$^{-1}$) = dry matter mass per plant × number of plants per hectare × nitrogen content of plants.

Nitrogen harvest index (NHI) = nitrogen accumulation of tuber / nitrogen accumulation of the whole plant.

Apparent nitrogen utilization ratio (%) = (nitrogen accumulation of plants treated with N fertilization-nitrogen accumulation of plants treated without N fertilization) / N fertilization amount × 100.

Nitrogen agronomic efficiency (kg kg$^{-1}$) = (yield of plants treated with N fertilization - yield of plants treated without N fertilization) / amount of nitrogen applied.

Nitrogen partial productivity (kg kg$^{-1}$) = yield / amount of nitrogen applied.

## Data analysis

All data were analyzed using one-way ANOVA in SAS 9.4 software (SAS Institute Inc., Cary, NC, USA). The analysis encompassed the determination of yam yield, morphological indicators, total sugar, starch, crude protein, total amino acid, moisture, and ash content, as well as N accumulation in tuber, stem, leaf, and bulbil, and N use efficiency. The least significant difference (LSD) was employed for multiple comparisons between treatment means. A mean difference was considered statistically significant when $P < 0.05$.

## Results and analysis

### Yield

As depicted in Fig 1, all fertilization treatments significantly increased yam yield compared to the control treatment (CK) ($P < 0.05$). However, the effects of different treatments varied. Notably, the combination of 25%OM + 75%CF exhibited the highest yield at 31.96 t ha-1, which was significantly superior to OM alone (22.72 t ha$^{-1}$), as well as the combinations of 50% OM + 50%CF (26.89 t ha$^{-1}$) and 75%OM + 25%CF (24.38 t ha$^{-1}$). Nevertheless, it did not demonstrate a significant difference when compared to CF alone (30.18 t ha$^{-1}$). The increase in

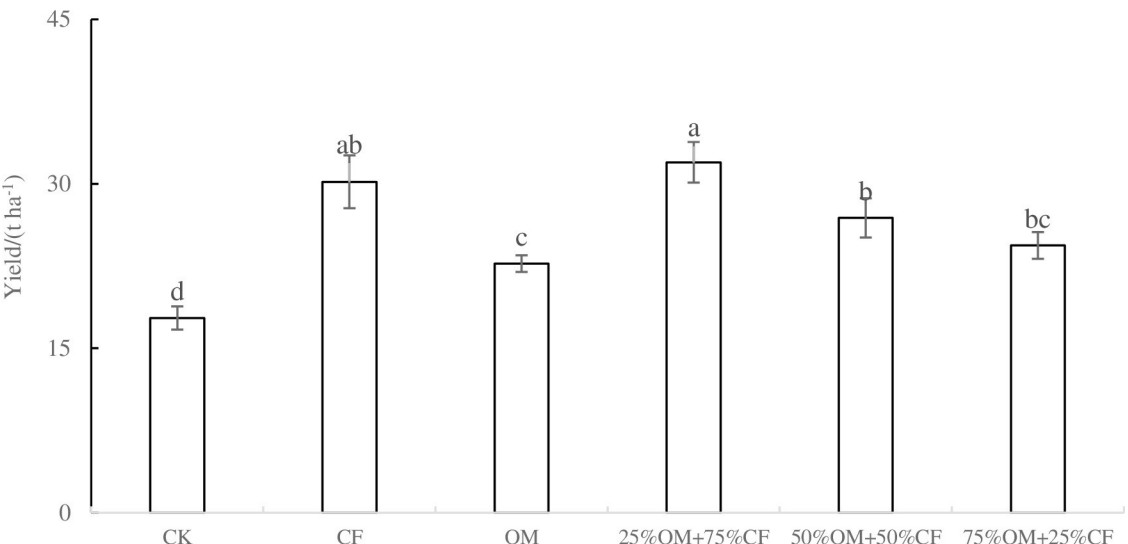

**Fig 1. Effect of partial substitution of chemical fertilizer with organic fertilizer on yam yield.** Note: Different letters indicate significant difference at $P < 0.05$ level.

yield achieved by OM was comparatively weaker and significantly lower than that observed with other fertilizer treatments.

## Morphological indexes

As shown in Table 2, fertilization significantly improved the morphological indexes of yam, although the effects varied among different fertilization treatments. 25%OM+75%CF resulted in the highest values for tuber length, tuber diameter, tuber fresh weight, and commercial rate, which were 69.23 cm, 5.51 cm, 1.15 kg, and 75.86%, respectively. There were no significant differences in tuber length and commodity rate among the different treatments, with values ranging from 63.48 cm to 69.23 cm and from 74.12% to 75.86%, respectively. However, the tuber diameter of OM was significantly lower than that of CF and 25%OM+75%CF (5.48 cm and 5.51 cm, respectively). The tuber fresh weight showed significant differences among different treatments, except for the non-significant difference of CF (1.11 kg) with 25%OM+75%CF (1.15 kg), as well as that of 75%OM+25%CF (0.82 kg) with OM (0.80 kg) and 50%OM+50%CF (0.91 kg). Significant differences were observed among the other treatments ($P < 0.05$).

**Table 2. Effect of partial substitution of chemical fertilizer with organic fertilizer on morphological indicators of yam.**

| Treatment | Tuber length /cm | Tuber thickness /cm | Tuber fresh weight /kg | Commodity rate /% |
|---|---|---|---|---|
| CK | 56.78± 1.08c | 4.66± 0.13c | 0.68± 0.07d | 60.87± 1.03b |
| CF | 67.12± 1.35a | 5.48± 0.14a | 1.11± 0.12a | 74.69± 1.77a |
| OM | 63.48±0.81 ab | 4.87± 0.07b | 0.80± 0.06c | 75.16± 2.38a |
| 25%OM+75%CF | 69.23± 2.17a | 5.51± 0.26a | 1.15± 0.13a | 75.86± 2.65a |
| 50%OM+50%CF | 65.31±1.79 ab | 5.03± 0.11ab | 0.91± 0.05b | 74.12± 1.14a |
| 75%OM+25%CF | 64.63±1.21 ab | 4.92±0.09 ab | 0.82±0.08 bc | 74.31±0.82 a |

Note: Different letters in the same column indicate significant level of difference P < 0.05

## Quality

The application of fertilizer has a significant impact on the quality of yam, as evidenced by substantial increases in total sugar, starch, crude protein, total amino acid and ash content (except for OM), along with a significant reduction in water content ($P < 0.05$). The contents of total sugar, starch, water and ash varied within ranges of 12.21% to 12.98%, 15.62% to 16.21%, 75.43% to 76.95%, and 0.66% to 0.77%, respectively among different fertilizer treatments; however no significant differences were observed between them except for partial substitution of chemical fertilizers with organic fertilizers which resulted in significantly higher levels of crude protein and total amino acids in yam compared to CF treatment at rates up to 25%OM+75% CF showing the highest increase at 18.40% and 29 .70%respectively (Table 3).

## Nitrogen accumulation

The distribution of nitrogen accumulation in different parts of yam follows the pattern of tuber > leaf > stem > bulbil, with proportions ranging from 68.81% to 73.76%, 15.74% to 17.73%, 8.35% to 8.72%, and 2.16% to 4.95%, respectively (Fig 2). In terms of total nitrogen accumulation, fertilization significantly increased the nitrogen accumulation in yam ($P < 0.05$). 25%OM+75%CF had the highest nitrogen accumulation at 156.91 kg ha$^{-1}$, followed by CF (153.56 kg ha$^{-1}$), with no significant difference between them. However, both were significantly higher than that of OM (137.78 kg ha$^{-1}$), 50%OM+50%CF (143.49 kg ha$^{-1}$), and 75% OM+25%CF (143.78 kg ha$^{-1}$).

## Nitrogen utilization

As presented in Table 4, the nitrogen harvest index of the control treatment (CK) was the highest, reaching 0.74. However, there were no significant variations between CK and other treatments, and the nitrogen harvest index (NHI) ranged between 0.69 and 0.74. 25%OM+75%CF had the highest nitrogen apparent utilization rate at 9.89%, followed by CF(9.09%). Both of these values were significantly higher than 75%OM+25%CF (6.76%), 50%OM+50%CF (6.69%) and OM (5.32%). Significant differences were observed in nitrogen agronomic efficiency among the different treatments. 25%OM+75%CF had the highest nitrogen agronomic efficiency at 33.93 kg kg$^{-1}$, followed by CF (29.68 kg kg$^{-1}$). 75%OM+25%CF and OM had the lowest nitrogen agronomic efficiency of 15.82 kg kg$^{-1}$ and 11.85 kg kg$^{-1}$, respectively. The nitrogen partial productivity was the highest for 25%OM+75%CF at 76.37 kg kg$^{-1}$, followed by CF (72.11 kg kg$^{-1}$). There were no significant differences between these two treatments, but both were significantly higher than 50%OM+50%CF (64.25 kg kg$^{-1}$), 75%OM+25%CF (58.26 kg kg$^{-1}$) and OM (54.29 kg kg$^{-1}$).

**Table 3. Effect of partial substitution of chemical fertilizer with organic fertilizer on yam quality.**

| Treatment | Total sugar /% | Starch /% | Crude protein /% | Total amino acids /% | Moisture /% | Ash /% |
|---|---|---|---|---|---|---|
| CK | 10.43± 0.28b | 14.01± 0.08b | 2.81± 0.07c | 1.74±0.03 c | 81.23± 1.68a | 0.61± 0.02b |
| CF | 12.21± 0.31a | 15.62± 0.11a | 3.37± 0.11b | 2.02± 0.05b | 76.95± 1.94b | 0.70± 0.01a |
| OM | 12.69± 0.24a | 15.66± 0.18a | 3.91± 0.16a | 2.59±0.06 a | 75.18± 1.79b | 0.66± 0.04ab |
| 25%OM+75%CF | 12.98± 0.33a | 16.21± 0.13a | 3.99± 0.15a | 2.62±0.09 a | 75.44± 2.14b | 0.77±0.03 a |
| 50%OM+50%CF | 12.77±0.29 a | 15.78± 0.17a | 3.86± 0.19a | 2.55± 0.02a | 76.25± 1.81b | 0.71± 0.05a |
| 75%OM+25%CF | 12.75±0.17 a | 15.71± 0.14a | 3.87± 0.12a | 2.52± 0.04a | 75.43± 1.26b | 0.67± 0.04ab |

Note: Different letters in the same column indicate significant difference at $P < 0.05$ level.

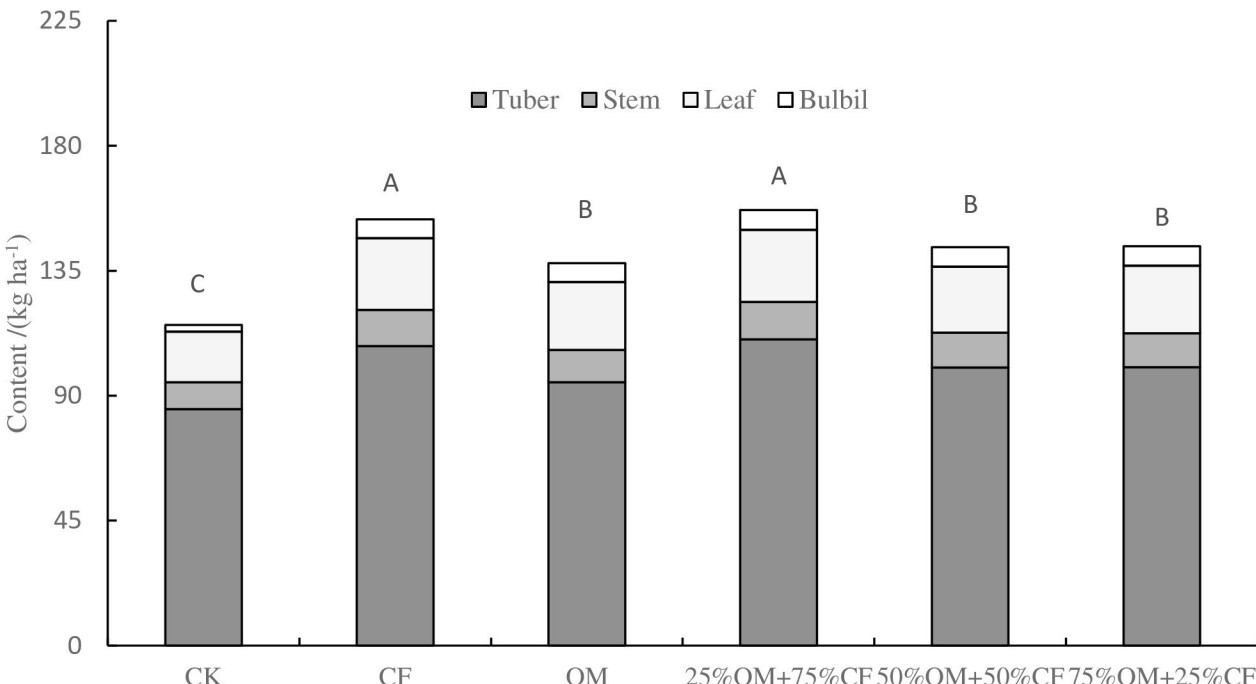

**Fig 2. Effect of partial substitution of chemical fertilizer with organic fertilizer on nitrogen accumulation in yam.** Note: Different letters indicate significant difference at $P < 0.05$ level.

## Discussion

It is worth emphasizing that achieving an optimal balance between organic and chemical fertilizers plays a pivotal role in maximizing nitrogen absorption, crop growth, and yield. This equilibrium not only enhances our understanding of nutrient dynamics in yam cultivation but also provides valuable insights for sustainable and high-yield agricultural practices on a broader scale. Compared to the conventional fertilizer treatment (CF), substituting 25% of chemical fertilizer with organic fertilizer resulted in a significant increase in yam yield. This could be attributed to the fact that the application of organic fertilizer improves soil fertility through the formation of organic-inorganic complexes and microaggregates, thereby increasing soil organic matter content [15]. Simultaneously, the applied organic fertilizer enhances the composition of the soil microbial community, expediting the decomposition and transformation of soil organic matter, thereby augmenting the continuous nutrient supply capacity of the soil [16]. However, when organic fertilizer replaced 50% of the chemical fertilizer, there was a

**Table 4. Effect of organic and chemical fertilizer application on nitrogen use efficiency of yam (mean ± standard deviation).**

| Treatment | NHI | Apparent utilization rate /% | Agronomic efficiency /(kg kg$^{-1}$) | Partial productivity /(kg kg$^{-1}$) |
| --- | --- | --- | --- | --- |
| CK | 0.74±0.01 a | — | — | — |
| CF | 0.70±0.03 a | 9.09±0.44 a | 29.68±1.79 b | 72.11±2.44 a |
| OM | 0.69±0.02 a | 5.32±0.51 b | 11.85±0.46 e | 54.29±2.19 c |
| 25%OM+75%CF | 0.70±0.01 a | 9.89±0.76 a | 33.93±1.35 a | 76.37±3.76 a |
| 50%OM+50%CF | 0.70±0.02 a | 6.69±0.48 b | 21.82±1.08 c | 64.25±5.48 b |
| 75%OM+25%CF | 0.70±0.01 a | 6.76±0.44 b | 15.82±1.37 d | 58.26±4.47 c |

Note: Different letters in the same column indicate significant difference at $P < 0.05$ level.

notable reduction in yield [17]. Prior research has suggested that an exclusive reliance on organic fertilizer as the sole nitrogen source can lead to yield decreases [18]. Consequently, it is crucial to consider the optimal blend of organic and chemical fertilizers. Under conditions of equal nitrogen supply, substituting 50% of the fertilizer's nitrogen with organic fertilizer can result in a slower release of nutrients and reduced ammonium and nitrate nitrogen content in the soil. As yam's nitrogen requirements intensify during later growth stages, these demands cannot be adequately met, ultimately leading to a significant decline in yield. This finding aligns with previous studies [19]. Our two-year fertilization experiment demonstrates the feasibility of partially replacing chemical fertilizer with organic fertilizer in yam cultivation in Guizhou Province. We conclude that maintaining a 25% ratio of organic fertilizer nitrogen to total fertilizer nitrogen preserves high tuber length, crude and fresh weight, as well as commercial rate in yam crops, ensuring consistent and substantial yields.

In the process of crop growth, fertilization exerts a significant impact on crop quality, second only to the effect of crop varieties [20]. Fertilization directly influences the physical, chemical, and biological properties of the soil, including the cycling of soil organic matter, nutrient supply, and microbial population structure. Consequently, once the crop variety is established, fertilization emerges as a pivotal agronomic practice that profoundly shapes crop quality [21]. Total sugar, starch, pivotal protein, and amino acid content serves as pivotal indicators for assessing the nutritional quality of yams. In comparison to exclusive chemical fertilization, the partial substitution of chemical fertilizers with organic alternatives augments significantly the levels of crude protein and total amino acids in yams. Chemical fertilizers promptly provide crops with readily available nutrients, Whereas organic fertilizers persistently supply nutrients persistently. Partial substitution of chemical fertilizers by organic fertilizers can meet the nutritional needs of yams at different growth stages, beneficially improving the quality of yams. Similar results have been observed in other crops such as peppers and tobacco [22, 23].

Accurate application nitrogen has been shown to be significantly enhance the intensity and distribution rates of nitrogen absorption in yam grains. The combined application of organic and chemical fertilizers has proven to "augment nitrogen accumulation in yam plants and notably elevate the nitrogen content in the stems [24, 25]. Our study reveals that the nitrogen accumulation and distribution ratio in yam stems significantly exceeded that of other plant parts, which is consistent with previous research findings [26]. Under conditions of isonitrogen fertilization, replacing 25% of isotopic fertilizers with organic fertilizer resulted in significantly greater total nitrogen accumulation throughout the entire plant and in the yam tubers compared to the conventional fertilizer treatment. Wu et al. [27], demonstrated 'N labeling technology, also illustrated that combining organic and inorganic fertilizers could enhance nitrogen accumulation in grains within both vegetative and reproductive plant organs of winter wheat, surpassing the effects of single applications of organic or inorganic fertilizers. Optimal accumulation and distribution of crop dry matter play a pivotal role in achieving high-quality and high-yield yam production. The combined application of organic and chemical fertilizers proves advantageous for optimizing water and nutrient conditions, promoting overall dry matter accumulation throughout the crop's growth cycle, as well as facilitating dry matter translocation to the stems [28, 29].

Nonetheless, as the proportion of organic organic fertilizer nitrogen relative to synthetic fertilize nitrogen increases, both the total nitrogen accumulation in plants and grains gradually decreases, aligning with previous research findings [30–33]. Additionally, it has been observed that as the proportion of organic fertilizer replacing chemical fertilizer increases, there is a corresponding rise in organic matter content. However, the mineralization process of organic matter exhibits relative sluggishness, resulting in a delayed release of nutrients. Consequently, the nitrogen derived from organic fertilizer may prove inadequate to fulfill the plant's nitrogen

uptake requirements, leading to insufficient absorption of available nitrogen and ultimately reduced nitrogen accumulation in the stems [34, 35].

The issue of low nitrogen utilization rates remains a prominent issue in Chinese crop production. Various indicators, including nitrogen harvest index, nitrogen apparent utilization rate, agronomic efficiency, and partial productivity, serve to assess nitrogen efficiency of nitrogen usage. Significant disparities were observed in terms of both agronomic efficiency and partial productivity of nitrogen fertilizer within the scope of this study. These findings ndicate an imbalance in the fertilization system in the region, which may be attributed to the application of nitrogen, phosphorus, and potassium or possibly excessive fertilizer use. Consequently, optimizing the fertilizer composition becomes imperative. Research has indicated that, under specific nitrogen application levels, reducing chemical fertilizer application while increasing organic fertilizer usage can substantially enhance yam's fertilizer utilization efficiency. It is crucial to emphasize that the optimization should target the organic fertilizer-to-fertilizer nitrogen ratio rather than merely seeking a higher ratio [36]. Li et al. [37] discovered that the peak agronomic use efficiency of nitrogen fertilizer in wheat was achieved when organic fertilizer replaced 20% of chemical nitrogen fertilizer. In our study, we observed a similar trend, where nitrogen fertilizer efficiency initially increased and then decreased as the proportion of organic fertilizer replacing chemical nitrogen rose. The highest nitrogen use efficiency for yams was attained when organic fertilizer replaced 25% of chemical nitrogen fertilizers. This finding is consistent with previous research results [38]. These findings emphasize the significance of adopting a nuanced approach to nitrogen management in yam cultivation, aiming to achieve an optimal balance between organic and chemical fertilizers for maximizing nitrogen utilization efficiency and overall crop productivity. The outcomes of this study offer valuable insights for promoting sustainable agricultural practices and contribute to enhancing the broader comprehension of nutrient management in crop production.

## Conclusion

Partial substitution of chemical fertilizers with organic fertilizers resulted in varying effects on yam yield, quality, and nitrogen utilization of *D. polystachya* as the substitution ratio increased. Substituting 25% chemical fertilizers with organic fertilizer demonstrated comparable or even superior performance to solely applying chemical fertilizers. Specifically, there were no significant differences observed between CF (chemical fertilizer) and 25%OM+75%CF (25% organic fertilizer + 75% chemical fertilizer) treatments in terms of yam yield, tuber length, thickness, fresh weight, commodity rate, total sugar content, starch content, moisture content, ash content,nitrogen accumulation amount,nigrogen harvest index, apparent utilization rate,and partial productivity. However,the latter treatment significantly increased the contents of crude protein and total amino acids as well as agronomic efficiency. Considering the economic benefits for farmers at this stage,it is recommended to simultaneously apply 25% organic fertilizer and 75% chemical fertilizer in the test area or similar regions.This approach ensures high yam yield while improving soil conditions..

## Supporting information

**S1 File.**
(XLS)

## Acknowledgments

Many thanks to Dr. Jianguo Huang at Southwest University and anonymous reviewers for their valuable comments on this manuscript.

## Author Contributions

**Data curation:** Yuxi Liu.

**Supervision:** Hua Zheng, Shuxia He, Qing Zhao.

**Writing – original draft:** Chao Sun.

**Writing – review & editing:** Hai Liu.

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
