## [Decision Letter · Decision Letter 0]

6 Sep 2023

PONE-D-23-21993

Partial Substitution of Chemical Fertilizer with Organic Fertilizer Increases Yield, Quality and Nitrogen Utilization of Dioscorea polystachya

PLOS ONE

Dear Dr. Liu,

Thank you for submitting your manuscript to PLOS ONE. After careful consideration, we feel that it has merit but does not fully meet PLOS ONE’s publication criteria as it currently stands. Therefore, we invite you to submit a revised version of the manuscript that addresses the points raised during the review process.

Please submit your revised manuscript by Oct 21 2023 11:59PM. If you will need more time than this to complete your revisions, please reply to this message or contact the journal office at plosone@plos.org. Please include the following items when submitting your revised manuscript:

We look forward to receiving your revised manuscript.

Kind regards,

Sofia Isabel Almeida Pereira

Academic Editor

PLOS ONE

Journal Requirements:

   "This work received funding from the Guizhou University of Finance and Economics 2021 introduction of talent scientific research start-up project (Grant No. 2021YJ051"

Additional Editor Comments:

The material and methods section must be carefully revised and improved. References must be added to the methods. Please revise the statistical analysis.

Captions of Tables and Figures should be self-explanatory. Please include the explanation of each treatment. 

The discussion section must be improved based on the reviewer's comments.  

Reviewers' comments:

Reviewer's Responses to Questions

**Comments to the Author**

1. Is the manuscript technically sound, and do the data support the conclusions?

Reviewer #1: Partly

Reviewer #2: Partly

2. Has the statistical analysis been performed appropriately and rigorously? 

Reviewer #1: No

Reviewer #2: No

3. Have the authors made all data underlying the findings in their manuscript fully available?

Reviewer #1: No

Reviewer #2: Yes

4. Is the manuscript presented in an intelligible fashion and written in standard English?

Reviewer #1: No

Reviewer #2: Yes

5. Review Comments to the Author

Reviewer #1: 1. The study has no novelty or authors should state the novelty

2. Authors are economical with major information needed in the materials n methods

3. Authors needs to include one more treatment i.e. 75%OM + 25%CF

4. Authors claimed to carry out correlation analysis in section 2.5 but there is no correlation matrix/table to show

5. See more corrections in the attached text

Reviewer #2: The manuscript provides valuable information. However, some points need to be clarified and improved. More description of the material used for vermicompost production is needed. The experimental design is deficient. The statistical analyses are incomplete. More discussion is needed. Some references are cited in the text. All observations are noted in the paper.

6. PLOS authors have the option to publish the peer review history of their article (what does this mean?). If published, this will include your full peer review and any attached files.

Reviewer #1: **Yes: **Aboyeji, C. M

Reviewer #2: No

---

## [Author Response · Author response to Decision Letter 0]

3 Nov 2023

Thank you very much for your positive comments on our manuscript entitled “Partial Substitution of Chemical Fertilizer with Organic Fertilizer Increases Yield, Quality and Nitrogen Utilization of Dioscorea polystachya” (PONE-D-23-21993). All the comments are valuable and helpful for improving our paper. We substantially revised our old version based on the reviewer’s comments. Thereafter, the manuscript was revised and edited by a native speaker and we hope the revised manuscript can meet the requirements. In this thoroughly revised version, changes in our manuscript within the document were highlighted by using red-colored text. Point-by-point responses to the reviewers are included, and the line number based on the attached file named “Revised Manuscript with track changes”.

---

## [Decision Letter · Decision Letter 1]

29 Nov 2023

PONE-D-23-21993R1Partial Substitution of Chemical Fertilizer with Organic Fertilizer Increases Yield, Quality and Nitrogen Utilization of Dioscorea polystachyaPLOS ONE

Dear Dr. Liu,

Thank you for submitting your manuscript to PLOS ONE. After careful consideration, we feel that it has merit but does not fully meet PLOS ONE’s publication criteria as it currently stands. Therefore, we invite you to submit a revised version of the manuscript that addresses the points raised during the review process.

We look forward to receiving your revised manuscript.

Kind regards,

Sofia Isabel Almeida Pereira

Academic Editor

PLOS ONE

Journal Requirements:

Additional Editor Comments:

Minor corrections:

Title

- replace "with organic fertilizer" by "by organic fertilizer.

Abstract

- Please rephrase this sentence - "Although there were no significant differences in tuber length and marketable yield among the fertilizer treatments, 25%OM + 75% CF exhibited the highest values at 69.23 cm and 75.86%, respectively, representing a 3.14% and 1.57% increase compared to CF. If there is not significant differences the authors could not state increases compared to control.

- Please rephrase the following sentences, as the infomration is repeated : The total sugar, starch, crude protein, total amino acid, and ash contents of D. polystachya significantly increased in all fertilizer treatments compared to the control (except for ash content between CK and OM). When compared to CF, the substitution of different ratios of organic fertilizer for chemical fertilizer increased the total sugar, starch, crude protein, total amino acid, and ash contents to varying degrees."

Line 137- The following sub-heading should be changed: " Measuring items and methods"? Not clear, please rephrase for clarity.

Line 177 - Please replace by "Data analysis"

Discussion

- Data regarding yam's quality are not discussed. Please imporve the discussion section.

The manuscript must be revised by a Native English speaker as it contains several grammatical faults.

Reviewers' comments:

Reviewer's Responses to Questions

**Comments to the Author**

1. If the authors have adequately addressed your comments raised in a previous round of review and you feel that this manuscript is now acceptable for publication, you may indicate that here to bypass the “Comments to the Author” section, enter your conflict of interest statement in the “Confidential to Editor” section, and submit your "Accept" recommendation.

Reviewer #1: All comments have been addressed

Reviewer #2: All comments have been addressed

2. Is the manuscript technically sound, and do the data support the conclusions?

Reviewer #1: Partly

Reviewer #2: Yes

3. Has the statistical analysis been performed appropriately and rigorously? 

Reviewer #1: N/A

Reviewer #2: Yes

4. Have the authors made all data underlying the findings in their manuscript fully available?

Reviewer #1: Yes

Reviewer #2: Yes

5. Is the manuscript presented in an intelligible fashion and written in standard English?

Reviewer #1: Yes

Reviewer #2: Yes

6. Review Comments to the Author

Reviewer #1: Kindly do some minor corrections as specified in the attached text nnnnnnnnnnnnnnnnnnnnnnnnnnnnnnnnnnnnnnnnnnnnnnnnnnnnnnnnnnn

Reviewer #2: The corrections and observations made in the first review were correctly modified, and some suggestions were also addressed.

7. PLOS authors have the option to publish the peer review history of their article (what does this mean?). If published, this will include your full peer review and any attached files.

Reviewer #1: No

Reviewer #2: No

---

## [Author Response · Author response to Decision Letter 1]

13 Jan 2024

1. Title

- replace "with organic fertilizer" by "by organic fertilizer.

Response: revised.

2. Abstract 

- Please rephrase this sentence - "Although there were no significant differences in tuber length and marketable yield among the fertilizer treatments, 25%OM + 75% CF exhibited the highest values at 69.23 cm and 75.86%, respectively, representing a 3.14% and 1.57% increase compared to CF. If there is not significant differences the authors could not state increases compared to control.

Response: We have rephrased this sentence.

3. - Please rephrase the following sentences, as the infomration is repeated : The total sugar, starch, crude protein, total amino acid, and ash contents of D. polystachya significantly increased in all fertilizer treatments compared to the control (except for ash content between CK and OM). When compared to CF, the substitution of different ratios of organic fertilizer for chemical fertilizer increased the total sugar, starch, crude protein, total amino acid, and ash contents to varying degrees."

Response: Rephrased.

4. Line 137- The following sub-heading should be changed: " Measuring items and methods"? Not clear, please rephrase for clarity.

Response: Revised. Changed into “Methods of index determination”.

5. Line 177 - Please replace by "Data analysis"

Response: Changed into "Data analysis".

6. Discussion

- Data regarding yam's quality are not discussed. Please imporve the discussion section. 

Response: We have added the yam quality discussion, and also improved the presentation.

7. The manuscript must be revised by a Native English speaker as it contains several grammatical faults.

Response: Revised.

---

## [Editor Report · Decision Letter 2]

18 Jan 2024

PONE-D-23-21993R2Partial Substitution of Chemical Fertilizer with Organic Fertilizer Increases Yield, Quality and Nitrogen Utilization of Dioscorea polystachyaPLOS ONE

Dear Dr. Liu,

Thank you for submitting your manuscript to PLOS ONE. After careful consideration, we feel that it has merit but does not fully meet PLOS ONE’s publication criteria as it currently stands. Therefore, we invite you to submit a revised version of the manuscript that addresses the points raised during the review process.

We look forward to receiving your revised manuscript.

Kind regards,

Sofia Isabel Almeida Pereira

Academic Editor

PLOS ONE

Journal Requirements:

Additional Editor Comments:==============================

**ACADEMIC EDITOR:**The manuscript needs to be revised by a native English speaker. Several grammar faults and unclear sentences are found throughout the text.==============================

---

## [Author Response · Author response to Decision Letter 2]

6 Mar 2024

Response: Thanks a lot for your suggestion. I have taken the necessary steps to enhance the linguistic quality of our manuscript. I engaged the assistance of a faculty member from the University of Florida, who provided valuable insights and refined the language throughout the paper.

---

## [Editor Report · Decision Letter 3]

12 Mar 2024

Partial Substitution of Chemical Fertilizer with Organic Fertilizer Increases Yield, Quality and Nitrogen Utilization of Dioscorea polystachya

PONE-D-23-21993R3

Dear Dr. Liu,

We’re pleased to inform you that your manuscript has been judged scientifically suitable for publication and will be formally accepted for publication once it meets all outstanding technical requirements.

Kind regards,

Sofia Isabel Almeida Pereira

Academic Editor

PLOS ONE
---

## [Editor Report · Acceptance letter]

1 Apr 2024

PONE-D-23-21993R3 

PLOS ONE

Dear Dr. Liu, 

I'm pleased to inform you that your manuscript has been deemed suitable for publication in PLOS ONE. Congratulations! Your manuscript is now being handed over to our production team.

Kind regards, 

on behalf of

Dr. Sofia Isabel Almeida Pereira 

Academic Editor

PLOS ONE